# Incidence of Peri-Implantitis and Relationship with Different Conditions: A Retrospective Study

**DOI:** 10.3390/ijerph19074147

**Published:** 2022-03-31

**Authors:** Víctor Astolfi, Blanca Ríos-Carrasco, Francisco Javier Gil-Mur, José Vicente Ríos-Santos, Beatriz Bullón, Mariano Herrero-Climent, Pedro Bullón

**Affiliations:** 1Department of Periodontology, School of Dentistry, Universidad de Seville, C/Avicena S/N, 41009 Seville, Spain; doctorvastolfi@gmail.com (V.A.); jvrios@us.es (J.V.R.-S.); bbullon@us.es (B.B.); pbullon@us.es (P.B.); 2Technological Health Research Center, Biomaterials of the Faculties of Medicine and Dentistry, International University of Cataluña, 08017 Barcelona, Spain; xavier.gil@uic.es; 3Porto Dental Institute, 4150-518 Porto, Portugal; dr.herrero@herrerocliment.com

**Keywords:** peri-implant disease, peri-implant mucositis, peri-implantitis, peri-implant health, risk factor, risk indicator, osseointegration, osseous defects

## Abstract

Articles on the prevalence of peri-implant diseases showed that 90% of peri-implant tissues had some form of inflammatory response and a prevalence of peri-implantitis from 28% to 51% according to various publications. Objective: To provide an overview of how risk factors can be related with peri-implantitis. Methods: A retrospective longitudinal study including 555 implants placed in 132 patients was evaluated based on the presence of peri-implantitis following the criteria of Renvert et al. 2018. Results: In total, 21 patients (15.9%) suffered peri-implantitis (PPG) and 111 patients (84.1%) did not suffer peri-implantitis (NPG). The results reveal that smokers have a high incidence of peri-implantitis (72.7%) compared to non-smokers (27.3%) (*p* < 0.0005). Another variable with significant results (*p* < 0.01) was periodontitis: 50% PPG and 23.9% NPG suffered advanced periodontitis. Systemic diseases such as arterial hypertension, diabetes mellitus, osteoporosis, and cardiovascular diseases do not show a statistically significant influence on the incidence of peri-implantitis. Patients who did not attend their maintenance therapy appointment had an incidence of peri-implantitis of 61.4%, compared to 27.3% in those who attend (*p* < 0.0001). From the results obtained, we can conclude that relevant factors affect peri-implantitis, such as tobacco habits, moderate and severe periodontitis, and attendance in maintenance therapy.

## 1. Introduction

Advances in implant dentistry have allowed implant treatment to become a common and important therapeutic resource in the replacement of missing teeth. Currently, dental implants have more than 95% success, and these results have been stable for many years [1]. Although most longitudinal studies have reported survival rates of around 90–95% for periods of 5–10 years, failures occasionally occur during implant treatment [2].

After the osseointegration process of dental implants, biological problems of infectious inflammatory origin can appear and affect the peri-implant tissues, this inflammatory process that occurs in implants is similar to that developed in natural teeth, infections have less resistance to destruction, mainly due to the lack of periodontal ligament [3].

Implants can be affected by two types of lesions. The first is inflammation around the peri-implant tissue without bone loss defined as mucositis. The second is peri-implantitis, also an inflammatory lesion that differs from mucositis because a marginal bone loss occurs and progresses diagnostically by radiograph [4].

The prevalence of peri-implant diseases is reported mainly by retrospective studies; authors such as Fransson et al., 2005, demonstrated that 90% of peri-implant tissues had some type of inflammatory response, while these authors found a prevalence of peri-implantitis in their studies of 28% [5]. Subsequently, Roos-Jansaker et al., 2006 reported a prevalence of peri-implant mucositis of approximately 48%, while 6.6% of the implants included in their study had peri-implantitis [6]. However, more recent studies such as those by Rodrigo et al. in 2018 show a prevalence of 51% of peri-implant diseases in the Spanish population [7].

The risk factors that generate these peri-implant diseases should be considered, with an increased prevalence in the case of tobacco use or patients with a history of periodontal disease, inadequate maintenance attendance, and oral hygiene being indicators of peri-implantitis. However, there are other factors with less evidence in the literature, such as medical history (diabetes, arterial hypertension, alcohol habits), the presence of at least 2 mm of keratinized mucosa, excess cement in rehabilitation, and prosthetic characteristics [8]. Therefore, peri-implant risk factors must be determined because they can influence the evolution and results of treatment [9].

The main objective of this article is to determine the incidence of peri-implantitis and to provide an overview of how the different risk factors identified in the study affect the increased incidence of peri-implantitis.

## 2. Materials and Methods

### 2.1. Study Design

The study was approved by the Macarena University Hospital Ethics Committee (Nº: 112018) on 14 December 2018, and follows the Declaration of Helsinki of 1975, later revised in 2013 [10]. All patients gave informed consent. The study was developed in the Dental School of the University of Seville (Seville, Spain), in the Academic Department of Periodontics and Implants in patients who had previously been treated in this area between 1 October 2006 and 30 June 2019.

In the first phase, the database of all patients treated at the Master of Periodontology and Implants of the Faculty of Dentistry of the University of Seville was reviewed to identify those who had received dental implants from October 2006 to June 2019 (1445 patients). The population was composed of male and female patients older than 18 years of age who received an individual cemented crown, an individual screw-retained prosthesis, a fixed screw-retained partial prosthesis, a fixed cemented partial prosthesis, partial prosthesis with cantilever, complete hybrid prosthesis, metal-ceramic prosthesis, and overdenture with two or four implants, or dental implants with at least 1 year of follow-up after prosthesis placement.

Subsequently, 445 patients were contacted by phone, all of whom had all the necessary data to be included in the study, with the aim of a clinical and radiological review of their implants. Many of them decided not to participate in the study due to different causes, death, continued treatment in other clinics, refusal to participate in the study, and radiographs from the day of implant placement not being available. With patients who agreed to attend their clinical examination and gave informed consent, a clinical and radiographic review was performed.

The final sample size was 132 patients with 555 implants and a mean age of 47 years (range between 18 and 84 years).

### 2.2. Oral Evaluation

The following clinical parameters and index were recorded for each implant at 6 sites (3 in vestibular and 3 in lingual) by a single examiner (VA).

Probing depth (PD) (in mm): to assess the depth of probing for each implant with a North Carolina periodontal probe applying a pressure of 0.15 Ncm [11].Modified Sulcular Bleeding Index (mBI): to assess bleeding or non-bleeding of each implant, registered from 0 to 3, according to the criteria of Mombelli et al. (BOP) [12].Modified plaque index (mPI): measure the amount of bacterial plaque on implants registered according to the O’Leary index, which represents the percentage of dental surfaces affected by bacterial plaque [13].Mucosal recession (REC): quantify the recession in each implant defined as the distance (mm) from the implant platform as a stable mark and mucosal margin [13].Suppuration (SUP): register the suppuration or not around implants according to the dichotomous scale (0/1), using the University of North Carolina probe 15, applying a force of 0.15 Ncm [11].

### 2.3. General Data

Sex: male, female.Smoking habits: No, Yes < 10/day, Yes > 10/dayAlcohol habits: No, moderate consumers (<0.5 L/day), heavy consumers (>0.5 L/day).Systemic diseases: arterial hypertension (AHT), diabetes mellitus (DM), osteoporosis, cardiovascular diseases.Periodontal disease: Non-, mild (level of clinical attachment—CAL 1–2 mm), moderate (CAL 3–4 mm), Advanced (>5 mm) [12].Maintenance: No, every 4 months, every 6 months, every 12 months.Reasons for tooth loss.Date of implant placement.

### 2.4. Implants Data

The implant was taken as the study unit to evaluate which type of prosthetic connection, diameter, and length of each implant have the most significant results in peri-implantitis.

Diameter and length of each implant.Type of prosthetic connection: 1: bone level, 2: tissue level, 3: external connection [14].

### 2.5. Diagnosis of Peri-implantitis

The authors classified peri-implantitis according to the criteria stipulated by Renvert et al., 2018 [8]:Evidence of visual inflammatory changes in the soft tissues of the peri-implant combined with bleeding on probing and suppuration [15].Increasing the depth of the probing pocket compared to the measurements obtained at the placement of the prothesis.Progressive bone loss in relation to the radiographic evaluation of the bone level assessment one year after implant prosthetic rehabilitation placement.

### 2.6. Radiographic Assessment

Standardized radiographs should be taken and compared with baseline radiographs when implants were placed in function [8]. The peri-implant radiographic bone loss (MBL) was determined by a calibrated examiner (VA) who calculated the possible bone loss by taking linear measurements from the most mesial and distal part of the implant platform to the crest bone in each implant. (Figure 1).

### 2.7. Satisfaction Questionnaire

Patients answered questions to subjectively evaluate the mastication, phonetics, aesthetics, and cleaning of implant-supported restoration and oral hygiene through the following question: ‘How do you evaluate the rehabilitations of your implants in terms of mastication, phonetics, aesthetics, and the possibility of cleaning: good, acceptable, or bad?’.

### 2.8. Statistical Analysis

Statistical analysis was conducted with SPSS (IBM Corp. Released 2017. IBM SPSS Statistics for Windows, Version 25.0. Armonk, NY, USA: IBM Corp). Descriptive analysis of all variables was performed, analyzing frequency and percentage. The variables were analyzed with a chi-square test to determine the groups that make a difference, and the standardized residuals were corrected according to the Haberman method. The statistical significance of the results was taken (*p* < 0.05); the lower the result, the greater the significance.

## 3. Results

The median of the follow-up period of the patients was 6.7 years (range 1 to 9 years). The incidence of patients with peri-implantitis was 21 (15.9%). No statistical differences were observed according to sex (male/female). Furthermore, the study results showed that in nonsmokers, only 12 patients (27.3%) have peri-implantitis compared to 18 patients (40.9%) who smoke less than 10 cigarettes per day and 14 patients (31.8%) that smoke more than 10 cigarettes per day (*p* < 0.0001). Another variable with statistically significant results is alcohol consumption (*p* < 0.05). Patients who do not consume (33 patients, 75%), have more significance of peri-implantitis than heavy consumers (5 patients, 11.4%), (Figure 2).

AHT, DM, osteoporosis, and cardiovascular diseases did not show statistical differences between PPG and NPG (Figure 3).

The results for the previous periodontitis were significant (*p* < 0.05) in patients with previous periodontal disease; 2 (4.5%) in healthy patients and increased to 22 (50%) for severe periods of periodontitis; 6 (13.6%) and 14 (31.8%) for mild and moderate states, respectively (Figure 4). Another study variable in that research is the reason for tooth loss; patients that lose their tooth due to periodontitis demonstrate statistically significant results; 37 (84.1%) patients with previous episodes of periodontal diseases also suffer peri-implantitis, *p* < 0.05 from 7 patients (15.9%) who do not have periodontitis. (Figure 4).

Patients who did not attend the periodontal maintenance show more peri-implantitis (61.4% PPG versus 27.3% NPG) than those that attend every 6 months (18.2% PPG versus 28.4% NPG) and those that attend every 12 months (18.2% PPG versus 43.2% NPG). Therefore, the frequency of periodontal maintenance has an influence on the peri-implantitis (Figure 5).

The parameters evaluated for the type of implant, diameter, and length do not show statistical differences with respect to diameter and length, and for the type of connection they were statistically significant (*p* < 0.0001). According to our results, the external connection is more likely to develop peri-implantitis compared to the internal connections at bone and tissue level implants. (Figure 6).

In general, all patients were satisfied after receiving implant treatment, and the results did not show significant differences in patients with peri-implantitis or healthy patients. Table 1.

## 4. Discussion

The main objective of this work is to determine the incidence of peri-implantitis in dental implants placed in patients of the Master in Periodontics and Implants of the University of Seville and to see the relationship between peri-implantitis and the different causal factors included in the study.

There should be a clear awareness of the increase in the prevalence of peri-implant diseases currently, with reference to the multicenter study by Rodrigo et al. in 2018 [7], suggesting that one of the two implants placed will develop peri-implant pathology. However, the results of the referenced studies indicate a need to improve the epidemiological reporting of scientific articles on peri-implant diseases, considering the risk factors that develop these biologic diseases.

Several studies published in the literature report a direct relationship between smoking and peri-implantitis. In the 2018 workshop published by Schwarz et al., smoking is mentioned as a risk factor for developing peri-implantitis, as in the results obtained in the present study [4]. Karoussis concluded in his study that 18% of peri-implantitis were in smokers, while only 6% were in non-smokers, these results are very similar to those obtained in the present investigation. Three cross-sectional studies confirmed these findings, reporting odds ratios of 32, 3, 5, respectively [16]. However, other authors who published longitudinal cross-sectional studies such as Koldsland et al. [17], Casado et al. [18], Dalago et al. [19] did not identify smoking as a risk factor for peri-implantitis. Therefore, no conclusive evidence has been reported in the literature that smoking is a risk factor for developing peri-implantitis. The reason for the contradictory data in the literature is mainly due to differences in the categorization of smokers and non-smokers, and most of the reported studies were based solely on the information provided by the patient for the assessment of smoking status. There is no conclusive evidence reported in the literature that smoking is a risk factor for developing peri-implantitis.

Regarding alcohol habits, the results of the present study show 75% peri-implantitis in non-drinkers (this is due to the large number of subjects studied), while in heavy drinkers (<0.5 L/day), these results are statistically significant (11.4%), demonstrating with these results the significant relationship between alcohol habit and the development of peri-implantitis. However, the results can be confusing because patients do not recognize their addiction to alcohol. Alcoholism has traditionally been associated with implant failure, mainly due to poor hygiene [20]. The results from the present investigation agree with the findings of Galindo et al. in 2005, where they state that the habit of alcohol consumption has a negative effect on peri-implant tissues, compromising the outcome of implant-prosthetic rehabilitations. However, more randomized controlled studies are needed to determine the relationship between alcohol and peri-implantitis.

For arterial hypertension (AHT), no statistically significant results were found in the present investigation in patients with arterial hypertension and peri-implantitis. There is a scarcity of bibliographic reports referring to hypertension and peri-implantitis, one of them is the one published by Wu et al. in 2016 [21]; they studied 325 implants and only obtained 0.6% of the complications. The difference in the results obtained in the present study from those published in the literature may be due to systemic control of patients in private medical centers; many of the patients selected in the study knew that they suffered from arterial hypertension but were not under medical treatment. It is known that antihypertensive drugs in general are beneficial for bone formation and bone remodeling and are associated with a lower risk of bone fractures; therefore, the use of antihypertensive drugs may be associated with a higher survival rate of osseointegrated implants. Most studies generalize about systemic conditions (other than diabetes) and conclude that it is a limited risk factor for developing peri-implantitis. Finally, the prevalence of hypertension is increasing in patients older than 60 years in developed countries. Patients with hypertension also require dental implant therapy, which is the reason that patients with AHT have an increase in peri-implant diseases. However, the impact of hypertension on the longevity of implant-prosthetic restorations is a topic for future research.

Since DM is one of the systemic diseases from which most dental patients suffer around 9.3% of the world’s population, the authors decided to add this variable to the present study. Nowadays, there are many publications that evaluate the relationship between DM and dental implants. In the study by Rokn et al. in 2017 [22], as in the present investigation, no relationship was found between DM and the incidence of peri-implantitis, the conclusion reached in Schwarz’s review [4] was that not all the articles they reviewed (12 reporting the relationship between DM and peri-implantitis) are conclusive as to whether diabetes is a risk factor for peri-implantitis. However, other authors such as Papi and Monje et al. [23,24] in their meta-analysis considered that DM is associated with an increased risk of peri-implantitis, but not with peri-implant mucositis, specifically Papi et al. reported that there is a 50% higher risk of detecting peri-implantitis in subjects with diabetes/hyperglycemia compared to non-diabetics [23].

Ultimately, the variability in the results may be due to the fact that, today, patients who come to the dental clinic are involved not only with their oral health but also with their systemic health, with better endocrinological controls, as well as regular glycemic monitoring and controls; however, although there is no conclusive evidence, there is a relationship between the presence of DM and peri-implantitis, and it is advisable for patients to keep their glucose levels under control at all times [25].

Patients with cardiovascular disease did not show statistical significance of the incidence of peri-implantitis (6.8%) than in healthy patients (87.5%). In most studies published in the literature, such as that of Renvert et al. in 2014 [26], factors related to peri-implantitis in a retrospective study show a higher risk of peri-implantitis in patients diagnosed with cardiovascular disease around 27.3%, which differs significantly from the results obtained by the authors, the main reason could be that the patients included in this study who had peri-implantitis had a mean age of 68.2 years, which was retrospectively evaluated, the authors obtained data on probing, depth, bleeding on probing, and radiographic bone levels of patients with dental implants. However, authors such as Lee et al. [27], with publications with less scientific evidence, agree with the results obtained in the present study, stating that they did not find a significant association between cardiovascular diseases and complications in dental implants.

Periodontal disease is one of the most diagnosed pathologies in the oral cavity. Recent studies carried out in the USA by Eke et al. [28] report that approximately 50% of the adult population (>30 years) suffer from periodontitis, in patients over 65 years of age it was 68%. The high incidence of this problem in the literature coincides with the results of the present study, 14 patients (31.8%) had moderate periodontal disease and 22 patients (50%) had advanced periodontal disease of a total of 132 patients, demonstrating with these results the statistical significance of peri-implantitis in periodontal patients. However, there is controversy in the literature on the possible association between a history of periodontitis (chronic or aggressive) and peri-implantitis. Authors such as Karoussis et al. [16] obtained peri-implantitis results in patients with previous periodontal disease of 31%, with results like those of the present study of 29% and 47% depending on the degree of involvement. Roccuzzo et al. [29] followed 101 patients who had dental implants placed, previously classified as not periodontally compromised, moderately compromised, and severely compromised. Although most publications are generally in agreement when examining the association between periodontitis and peri-implantitis, it should be noted that there are also contradictory literature references, such as the studies published by Schwarz et al. 2017 [30], Marrone et al. 2013 [31] evaluated 103 patients with implant-prosthetic restorations with a minimum follow-up period of 5 years where they did not obtain a statistically significant relationship between a history of periodontitis and peri-implantitis, other authors such as Rokn et al. 2017 in a cross-sectional study of 134 patients, could not demonstrate an increased risk of peri-implantitis in patients with a history of periodontitis [22]. Ultimately, the controversy in the results in the different studies may be due to variability in case definitions for a history of periodontitis, peri-implantitis; therefore, there is strong evidence from longitudinal studies and cross-sectional studies that a previous history of periodontitis constitutes a risk factor, indicative of peri-implantitis.

With respect to maintenance therapy carried out by patients with dental implants, the results obtained in the present investigation indicate a statistically significant difference in that patients who do not attend maintenance have a higher incidence of developing peri-implantitis; however, patients who do attend maintenance have a lower incidence. Classic studies on periodontal disease have shown that inadequate maintenance is associated with increased tooth loss [32]. Several studies have been published in the literature on the possible association between poor plaque control, lack of regular maintenance, and peri-implantitis. The results obtained from a longitudinal study involving patients diagnosed with mucositis, indicating the importance of plaque control in the prevention of peri-implantitis [29], obtained results similar to those of the present study, with a lower incidence of peri-implantitis over a 5-year period in patients undergoing maintenance therapy (18%). These results are in agreement with those published by authors such as Roccuzzo et al. [28], who reported that patients who were not included in maintenance therapy over a 10-year period developed peri-implantitis (41%), while those who attended maintenance visits decreased the incidence by up to 27%. In contrast, contradictory data have also been reported, where different cross-sectional studies such as those published by Roos-Jansaker et al. [33] and Dvorak et al. [34] showed that there was no correlation between bacterial plaque values with their respective maintenance and peri-implantitis, but it is definitely scientifically supported that regular lack of maintenance is a risk factor for developing peri-implantitis; however, more randomized controlled studies are needed in the future.

## 5. Conclusions

From the results obtained in our study, we can conclude that there is a significant association between tobacco habits (a higher number of cigarettes, greater incidence) and the possibility of developing peri-implantitis.

Another conclusion derived from the results achieved in the investigation is that it is not statistically significant that patients with a history of systemic diseases such as hypertension and diabetes mellitus have a higher risk of developing peri-implantitis compared to healthy patients.

Patients with mild, moderate, or severe periodontitis have more risk of suffering from peri-implantitis compared to patients without periodontitis. Furthermore, if the reason for tooth loss was a previous history of periodontal disease, the risk of peri-implantitis increases, compared to if the reason for tooth loss was caries or trauma.

Patients who do not attend maintenance therapy will have a higher risk of developing peri-implant disease.

Finally, all of these factors are framed on an individual patient basis, whose situation needs to be analyzed on a case-by-case basis; therefore, more randomized controlled studies will be needed to develop future research.

## Figures and Tables

**Figure 1 ijerph-19-04147-f001:**
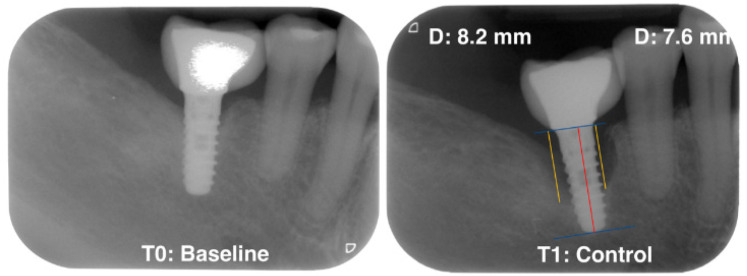
Measurement of radiographic bone loss.

**Figure 2 ijerph-19-04147-f002:**
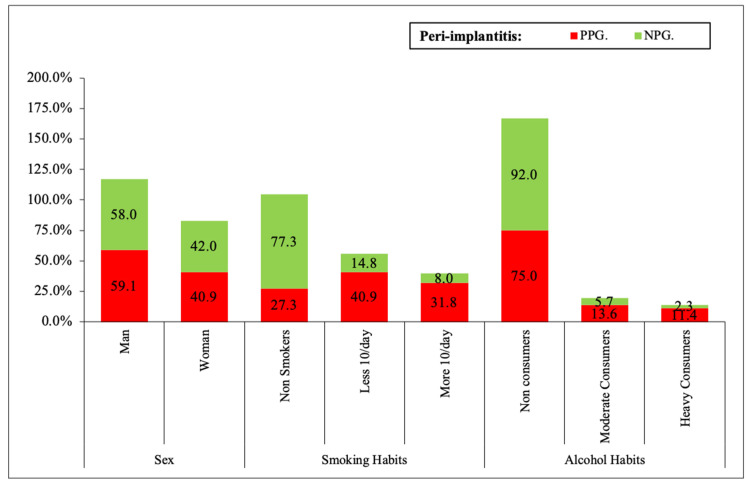
Statistical significance of general variables in peri-implantitis taking the patients as unit. Different variables: PPG—Non-smokers (27.3% * 5), NPG—Non-smokers (77.3% * 5). PPG—less than 10/day (40.9% * 3), NPG—less than 10/day (14.8% * 3). PPG—More 10/day (31.8% * 4), NPG—More 10/day (8% * 4). PPG—Non-Consumers (75% * 2), PPG—Non-Consumers (92% * 2). PPG—Heavy consumers (11.4% * 1), NPG (2.3% * 1). * 1: *p* < 0.05; * 2: *p* < 0.01; * 3: *p* < 0.001, * 4: *p* < 0.0001; * 5: *p* < 0.00001.

**Figure 3 ijerph-19-04147-f003:**
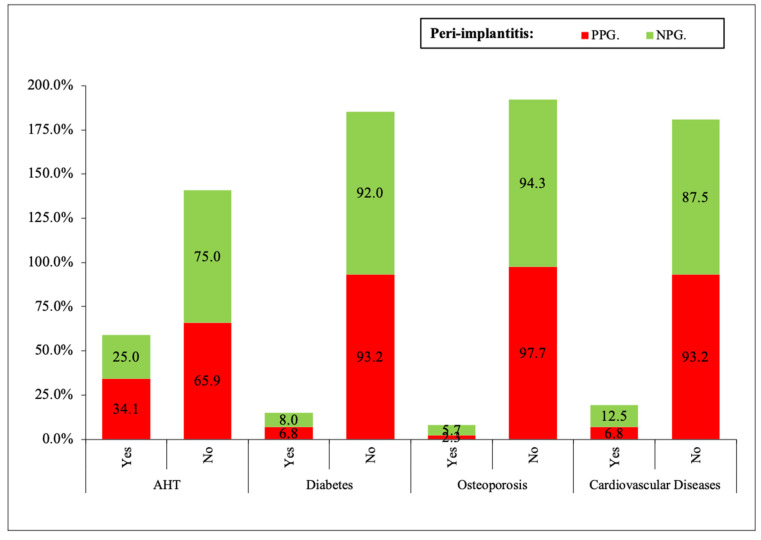
No statistical significance of systemic diseases in peri-implantitis taking the patients as unit.

**Figure 4 ijerph-19-04147-f004:**
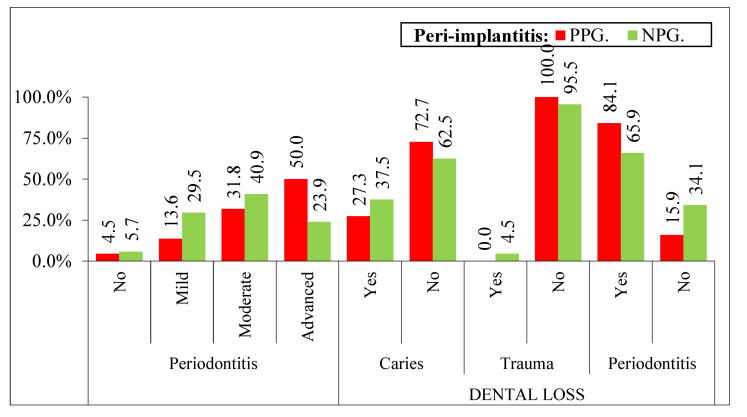
Variables related to periodontal disease and tooth loss according to peri-implantitis taking the patients as the unit. Different variables: PPG—Non-Periodontal (4.5% * 1), NPG—Non-Periodontal (5.7% * 1). PPG—Mild Periodontitis (13.6% * 1), NPG—Mild Periodontitis (29.5% * 1), PPG—Periodontitis (84.1% * 1), NPG—Periodontitis (65.9% * 1). PPG—no periodontitis (15.9% * 1), NPG—no periodontitis (34.1% * 1). 1: *p* < 0.05.

**Figure 5 ijerph-19-04147-f005:**
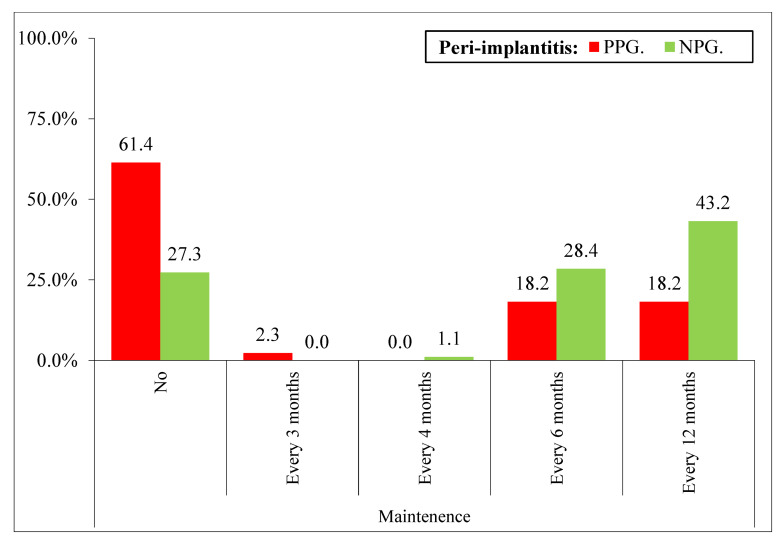
Variables related to follow-up and maintenance according to peri-implantitis taking the patients as the unit. Different variables: PPG—Non-maintenance (71.4% * 3), NPG—Non-maintenance (27.3% * 3), PPG—Maintenance every 12 months (18.2% * 2), NPG—Maintenance every 12 months (43.2% * 2). * 1: *p* < 0.05; * 2: *p* < 0.01; * 3: *p* < 0.001, * 4: *p* < 0.0001, * 5: *p* < 0.00001.

**Figure 6 ijerph-19-04147-f006:**
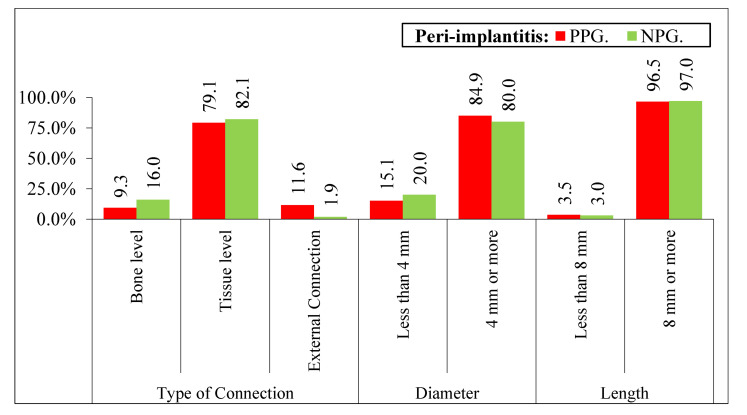
Qualitative variables related to the implant according to peri-implantitis with the implant as a unit. Different variables: PPG—External Connection (11.6% * 5), NPG—External Connection (1.9% * 5). * 1: *p* < 0.05; * 2: *p* < 0.01; * 3: *p* < 0.001, * 4: *p* < 0.0001, * 5: *p* < 0.00001.

**Table 1 ijerph-19-04147-t001:** Qualitative variables of patients’ satisfaction related to peri-implantitis. Signification of different variables: * 1: *p* < 0.05; * 2: *p* < 0.01; * 3: *p* < 0.001, * 4: *p* < 0.0001 and * 5: *p* < 0.00001.

Variable	Categories	Yes	No	Signification
Mastication	Good	67.4	75.9	**<0.05**
	Acceptable	12.8	13.9	
	Bad	**19.8 * 1**	**10.2 * 1**	
Phonetic	Good	72.1	67.6	**<0.05**
	Acceptable	**17.4 * 1**	**27.7 * 1**	
	Bad	**10.5 * 1**	**4.7 * 1**	
Aesthetic	Good	**40.7 * 4**	**64.8 * 4**	**<0.001**
	Acceptable	**44.2 * 3**	**26.9 * 3**	
	Bad	**15.1 * 2**	**8.3 * 2**	
Hygiene	Good	**58.1 * 2**	**72.9 * 2**	**<0.0001**
	Acceptable	19.8	20.9	
	Bad	**22.1 * 5**	**6.2 * 5**	

## Data Availability

The data presented in this study are available on request from the corresponding author. Data are not publicly available due to privacy.

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
