# Peer review of "Incidence of Peri-Implantitis and Relationship with Different Conditions: A Retrospective Study"

_ijerph, 2022, doi:10.3390/ijerph19074147_

Round 1

Reviewer 1 Report

Dear authors, the paper is interesting , but some important changes should be made in order to be published, I attached pdf with suggestions.

Author Response

Dear Reviewer 1,

Thank you very much for your time, all your suggestions have been taken into account.  

The paper has been improved, and we hope that now follow your indications in order to be published.

  • The grammatical mistakes that was mentioned at the pdf have been corrected.
  • Figure 1 has been deleted, hardly concern the objective of the paper.
  • Figure 2 has been modified, we expect that now is adequate.
  • The specific criteria of inclusion and exclusion of the study for the participants have been detailed.
  • The reference to the Declaration of Helsinki has been added.
  • Some parts of the text weren’t located in the corresponding section, it has been solved.
  • Diagram, figures, tables and bibliography order have been revised.
  • Statistical analysis of the sample size has been added.
  • The conclusions have been summarized, to be more concise and concrete.

Reviewer 2 Report

This study reports a cross sectional evaluation of a retrospectively defined cohort.

The level of evidence is poor, because of many factors.

In the list below, some of them are reported:

  • confused and disorganized reporting. The article is extremely verbose, and many paragraphs are in the wrong section; questionnaires are reported in the body text instead of tables.
  • confused analysis. The sentence “The sample size was determined to be 555 to obtain the most accurate final results. To reflect a true population value, with sensitivity and specificity test, confidence intervals was made for all values, with a power of 80%, probability of error less than 5%” is not clear, since the sample size was not calculated, and sensitivity and specificity test refer to diagnostic tests
  • the analysis uses implants as units even for systemic factors. It is not acceptable to make an implant-based analysis of factors as sex or systemic disease since different implants in the same mouth can’t have different sex or systemic disease. This is a spurious increase of the sample size, that could jeopardize the results.
  • conclusions do not match the aims of the study and “Patients with a history of systemic diseases” and other sentences refer to patients, while implant was inexplicably used as unit, showing both that the poor analysis and the inconsistency of the study

Behind this study there is clearly a huge work, and for sure the scientific community needs the most data as possible on peri-implantitis. Unfortunately, in its current form, the study cannot be published.

I suggest to the authors is to re-run a proper analysis and to rewrite the paper following the current scientific writing paradigm.

Author Response

Dear reviewer 2,

-  This study is a cross sectional evaluation of a retrospectively defined cohort. We know that is not the most relevant type of study but it is in the top of the base of the scientific pyramid of evidence, making it a valid study to be published.
- Authors have improve the evidence of the paper. Paragraphs of the text has been organized properly, rewriting parts of it or summarizing when it was necessary. Figures, tables and bibliography has been added and modified.
- The sample size was calculated, and the description have been added in section "2.1. Study Design".
- Following the instructions of the reviewer, to have stronger results, the statistics have been recalculated, changing the unit from implant to patient.
- The conclusions have been modified, hoping that the reviewer will find them appropriate. 
The reviewer´s suggestion has been taken into account, and the article has been rewritten to improve all aspects. Authors hope that the new approach make it liking of the reviewer for publication.   

Round 2

Reviewer 1 Report

Dear Authors I would need the final version with the last modification to be checked

Reviewer 2 Report

 Accept in present form